# The Gifted Rating Scales-Preschool/Kindergarten Form (GRS-P): A Preliminary Examination of Their Psychometric Properties in Two Greek Samples

**DOI:** 10.3390/diagnostics12112809

**Published:** 2022-11-15

**Authors:** Maria Sofologi, Georgia Papantoniou, Theodora Avgita, Aikaterina Lyraki, Chrysoula Thomaidou, Harilaos Zaragas, Georgios Ntritsos, Panagiotis Varsamis, Konstantinos Staikopoulos, Georgios Kougioumtzis, Aphrodite Papantoniou, Despina Moraitou

**Affiliations:** 1Laboratory of Psychology, Department of Early Childhood Education, School of Education, University of Ioannina, 45110 Ioannina, Greece; 2Institute of Humanities and Social Sciences, University Research Centre of Ioannina (U.R.C.I.), 45110 Ioannina, Greece; 3Laboratory of Art, Motor Expression and Didactic Application, Department of Early Chidlhood Education, School of Education, University of Ioannina, 45110 Ioannina, Greece; 4Laboratory of Neurodegenerative Diseases, Center for Interdisciplinary Research and Innovation (CIRI—AUTH) Balkan Center, Buildings A & B, Aristotle University of Thessaloniki, 57001 Thessaloniki, Greece; 5Department of Informatics and Telecommunications, School of Informatics and Telecommunications, University of Ioannina, 47150 Arta, Greece; 6Department of Hygiene and Epidemiology, School of Medicine, University of Ioannina, 45110 Ioannina, Greece; 7Department of Educational and Social Policy, University of Macedonia, 54636 Thessaloniki, Greece; 8Department of Computer, Informatics and Telecommunications Engineering, International Hellenic University, 62124 Serres, Greece; 9Department of Turkish Studies, National and Kapodistrian University, 10559 Athens, Greece; 10Laboratory of Psychology, Section of Experimental and Cognitive Psychology, School of Psychology, Aristotle University of Thessaloniki, 54124 Thessaloniki, Greece

**Keywords:** gifted rating scales, preschool children, intelligence, executive functions, giftedness

## Abstract

The present paper is based on data of two samples concerning the Gifted Rating Scales-Preschool/Kindergarten Form (GRS-P) that aimed to gain insight into the psychometric properties (internal consistency reliability, structural and convergent validity) of the Greek version of the GRS-P. In both studies, teachers estimated their students’ giftedness with the GRS-P and executive functions with the Childhood Executive Functioning Inventory (Study 1). In Study 2, kindergarteners were examined in cognitive measurements which included the colored progressive matrices, the children category test, the Athena test, and the mini-mental state examination. Statistical analyses (EFA, CFA, Cronbach’s *α*, and Pearson’s *r* coefficients) revealed the excellent internal consistency of the scales as well as their good factorial and convergent/discriminant validity. In relation to the children’s cognitive ability measures, it emphasized the fact that the GRS-P is a reliable and valid tool for teachers to assess their gifted students in a Greek cultural context.

## 1. Introduction

### 1.1. Evaluation of Giftedness

During the last two decades, increasing research attention has been given to the evaluation of gifted and high-ability students. The rapidly changing modern society has a great need for superior achievers. Gifted children have been characterized by most societies as human resources which contribute to culture in many ways. Research about gifted and highly talented students points to the great diversity among this heterogeneous group [1]. Furthermore, there has been existing difficulty in finding one single research-based definition to describe the diversity of the gifted and talented population, and the number of overlapping definitions of giftedness proposed in educational research mirrors the complexity of accurately defining giftedness. This specific issue underlies the significant difficulty and complexity of defining the phenomenon of giftedness with certainty [2].

More recently, in the sphere of education, the research community emphasizes that there is growing attention on educating talented learners who will be tomorrow’s leaders [3]. Additionally, an essential step in the scientific advancement of giftedness is the recognition of high-potential students more accurately and efficiently, a research idea that mirrors the heterogeneity of this group [4]. Historically, giftedness has been conceptualized based on performance on intelligence tests. More specifically, at the beginning of the century, when intelligence was synonymous with giftedness, intelligence assessment instruments and performance tests dominated the processes of recognizing giftedness [3,5]. Exceptional general intelligence was once thought to be the hallmark of giftedness but now is regarded as only one manifestation of giftedness [6].

To evaluate academically talented students, theorists created two research frameworks that can be grouped to (a) the assessment of students’ performance with screening tests available to complement the IQ test in providing a more comprehensive picture of their cognitive abilities [7]; and (b) alternative assessment methods [8]. The first research framework incorporates different assessment instruments for evaluating the child’s cognitive ability and intelligence [9]. These are standardized intelligence tests widely administered [10,11]. While measures of intelligence or cognitive abilities are widely used in the assessment of giftedness, many authors have issued cautions regarding the application and interpretation of scores for this purpose. Such issues include the possible depression of scores from ceiling effects, the cultural loadings, the arbitrary choice of cutoff scores, the inclusion of processing speed in many of the measures, and the uneven profile of abilities found in gifted students [12,13]. In addition, measures of intelligence or cognitive ability may underestimate the potential of highly creative children who provide divergent-type responses on tests that reward the one “right” answer. Recently, nonverbal ability tests have gained attention in helping to evaluate gifted populations. One important reason is that many educators contend that nonverbal ability tests are more fair or equitable for culturally or linguistically diverse populations [14]. The logic underlying this view is that many children are intellectually gifted but not able to demonstrate high academic achievement because of unequal opportunities to learn in essence, they have been exposed to a substantially diminished, understimulated, and/or markedly culturally different educational environment in their early development. Nonverbal ability tests are measures of general ability. Although some nonverbal tests are quite comprehensive and include multiple subscales, they are intended to measure the various abilities underlying intelligence, and not multiple types of giftedness (e.g., artistic ability, leadership, science, creative writing) [14]. On the other hand, Sternberg (1982) [15] pointed out that test situations can be highly anxiety-provoking for some students, that the tests are measuring not just ability or intelligence but also previous learning or achievement, and that precise scores do not necessarily reflect valid scores. Despite their weaknesses, when used with care and consideration, in the context of an assessment that includes multiple sources of information, measures of intelligence or cognitive ability have the potential to provide valuable information in the gifted identification process [13]. Additionally, a recent trend in assessment has been an interest in developing a means for practitioners to effectively use a cross-battery assessment approach to testing. Specifically, a multiple assessment approach allows the practitioner to identify the broad and narrow factors they have assessed using their primary instrument and supplement using subtests from other assessment batteries [16]. For example, according to Duggan and Garcia-Barrera [17], executive function and intelligence are constructs that consist of common conceptual elements. The proper functioning of the EF is associated with high IQ. Problem-solving and insight concepts related to EF correspond to the behaviors of gifted individuals. According to the findings of Duff, Schoenberg, Scott, Russell, and Adams [18], there is a strong relation between executive functioning and working memory capacity as measured by standardized neuropsychological tests. Their analyses indicated that the two cognitive domains shared more than 50% of variance, revealing that intellectual functioning is strongly related to both executive functioning and memory [19,20,21]. Furthermore, in their empirical review, Foley et al. [22] concluded that it would be important to include measures of executive function in assessments, with the consideration that gifted students may not show clinical levels on these measures to the same degree as the general population. A review of the literature suggests that benchmarks for executive functioning could facilitate a better understanding of gifted populations [23].

The second research framework incorporates several teachers’ and parents’ rating scales based on their personal judgments and estimations to evaluate giftedness, and extends giftedness, beyond IQ scores, considering that ratings by teachers and parents are still considered a significant tool for identifying highly talented students [3,24].

Based on a large body of research, such assessment approaches enable the identification of elements or characteristics of one’s potential (such as motivation, creativity, leadership, and interest in a subject), which are not perceived by most objective cognitive measurements [25,26]. Accordingly, the literature review reveals promising research data concerning specific scales that assess teachers’ estimations as a screening instrument to help identify high-level potential or gifted students. Teacher rating scales are widely used in the screening and identification of students for participation in programs for gifted students. They are possibly the most frequently used identification tools, coming after intelligence testing, in assessing gifted potential [5]. In specific, obtaining teacher opinions as a way of identifying gifted students has become a widely used method in recent years [27]. The benefits of teacher observations over parents in identifying gifted children have been well documented, and it has been found that the benefit of observations of teachers and families to determine the gifted children is of vital importance. This is true, especially in recent years where it has been emphasized as being significant in determining gifted children, due to the fact that the teacher assessment scales are being developed and that the observations of teachers are the most detailed ones carried out in the long term [28]. Identification protocols employ teachers as informants for superior learners [3,28,29]. Indicative scales for teachers or parents that have been used to assess giftedness in young children include the Scale for Rating the Behavioral Characteristics of Superior Students (SRBSS and SRBCSS-R) [30]. This is the most common scale teachers and parents use to highlight gifted students. Other assessment approaches for giftedness are the Gifted and Talented Evaluation Scale (GATES) [29], the Scales for Identifying Gifted Students (SIGS) [31], the HOPE Scale (the HOPE Scale) [32], and the Gifted Rating Scales (GRS) [33]. Moreover, although a lot of published teacher-gifted rating scales are used in schools, there is limited evidence to support their technical adequacy and clinical usefulness [23]. Therefore, Pfeiffer and Jarosewich [33] designed the Gifted Rating Scales (GRS) to meet the need for a teacher rating scale with acceptable psychometric properties to be used as a technically and diagnostically appropriate instrument as part of the gifted students’ identification process.

### 1.2. The Gifted Rating Scales

The Gifted Rating Scales [33] are teacher evaluation scales, which are based on a multidimensional model of giftedness, and are designed to assess the characteristics of a gifted profile along with a wide range of ages. In specific, the Gifted Rating Scales [33] are based on the Munich Model for the Identification of Giftedness [34,35], and measure different facets of giftedness, not just academic and intellectual abilities. According to this model, giftedness arises in the areas of intellect, creativity, social competence, artistic (musical) ability, and psychomotor ability. The individual potentials of giftedness correspond to particular academic or nonacademic achievement areas. In addition to cognitive abilities, various (noncognitive) personality characteristics such as motives, interests, self-concepts, and so on, are involved. Family and school socialization factors are important learning environmental conditions for developing expertise and domain-specific performances [35]. As a result, the importance, and the usefulness of accurate identification of high-achieving students are the most critical issue that gifted research circles must illuminate. More recently, researchers have also started paying more attention to traits such as motivation, leadership, creativity, and artistic talent [36]. They support a broader-based conception of highly talented or privileged children which is based on an amalgamation of nonintellectual qualities and intellectual potentials, such as motivation, self-concept, and creativity [37]. In specific, the GRS consists of a Preschool/Kindergarten Form (GRS-P) for ages 4.0 to 6.11 and a School Form (GRS-S) for ages 6.0 to 13.11 [33], and was designed to be reliable, and valid. The GRS-P consists of five scales (namely: intellectual, academic, creativity, artistic, and motivation) with 12 items each (60 items). The items of the GRS-P represent skills and behaviors developmentally appropriate for preschool and kindergarten students, whereas the items of the GRS-S reflect more developmentally advanced skills or behaviors. The GRS-S Form consists of six scales with 12 items each for a total of 72 items. The sixth scale is leadership, which is not included in the GRS-P. Both standardization samples were stratified to match the latest U.S. census in terms of ethnicity, parent education level, and regional representation [33]. More specifically, Cronbach’s alpha internal consistency ranged from 0.98 to 0.99 for the five GRS-P scales and ranged from 0.97 to 0.99 for the six GRS-S scales [33]. The findings of high-reliability coefficients corroborate those reported in the Chinese version [38], the Puerto Rican version [39], and the Arabic Version of the GRS-S [40]. As regards the five GRS-P scales, Karadag and Pfeiffer (2016) [41] have also found Cronbach’s alpha to range from 0.96 to 0.98 in a Turkish sample, and Siu in a Chinese sample. Coefficient alphas ranged from 0.86 to 0.93 [42].

However, the information, which is presented in the manual [33], does not support the two GRS forms’ structural validity, and taking into account that no overall score is available on both of them, the proposed (in an item-level analysis) theoretical structure of the GRS-P and the GRS-S consists of 5 and 6 first-order constructs (latent variables), respectively [42,43]. Although Pfeiffer et al. [33] stated that they conducted factor analyses to examine the internal structure of the GRS, they did not specify, in the manual [33], which kind of factor analyses were conducted [42], while they have presented this kind of results from item-level analyses supporting the proposed 6-factor structure as regards mainly the GRS-S [44,45,46].

To our knowledge, the proposed theoretical structure of the GRS-P consisting of 5 first-order constructs (latent variables) has been tested, with item-level analyses (CFA), by Karadag and Pfeiffer [41] in a Turkish sample. However, it is noteworthy that the indices of their proposed verified model ranged from not accepted (RMSEA = 0.070) to marginally accepted (CFI = 0.919) [44,45,46]. Furthermore, although the possibility that a general factor could also account for most of the variance captured by GRS ratings cannot be rejected, the GRS-P scoring model has been examined using a bifactor model approach only by Benson and Kranzler [47] in the standardization sample in the norming of the GRS-P. According to Benson et al. [47], the aforementioned possibility is reinforced by the mean scale intercorrelations across age groups for the GRS-P and GRS-S (0.80 and 0.74, respectively), which show high correlations among the scales and indicate a need to investigate whether one or more latent variables (first-order factors) could account for the shared variance among the 5 and 6 proposed scoring structures (as observed variables) of the GRS-P and GRS-S, respectively.

Regarding evidence of convergent validity, which is indicated when tools designed to measure the same construct correlate highly, the manual [33] referred to several studies in which scores on the GRS were uniformly positively correlated with external criteria (including intelligence, achievement, creativity, artistic talent, motivation, and leadership). In specific, as regards intelligence tests, low and moderate positive correlations between GRS-P scale scores and WPPSI-III scores were statistically significant at the *p* < 0.001 level, showing a general correlation between GRS-P ratings of giftedness and students’ overall intellectual ability. Additionally, low, and moderate positive correlations between GRS-S and scale scores on WISC-III were statistically significant at level 0.001 or lower (*p* < 0. 001).

In summary, an examination of the correlations between GRS scores and scale scores on WPPSI-III and WISC-IV reveals overall convergent validity in that the intellectual and academic ability scales of both GRS-P and GRS-S correlate most significantly with scale scores of the WPPSI-III and WISC-IV. Additionally, research findings indicate that the examination of the correlations between the scores of the two GRS and the scores of the total IQ indices, as well as the individual scores on the WPPSI-III, WISC-IV, and WIAT-II subtests, reveal the existence of high convergent validity. The existence of this similar pattern is likely due to the high internal correlation observed between the intellectual and academic scales [33].

From the aforementioned statements, it is a clear fact that the concept of giftedness is too broad and prone to controversy when attempting a definition [48]. Intelligence, and in particular IQ, is traditionally assumed to be the most decisive variable in the definition of giftedness; however, several researchers, as stated above, point out its ambiguity and insufficiency for this definition, including other variables related to motivation, personality, or creativity in the explanation of high capabilities and high performance [49,50]. This spectrum of the variables involved in giftedness highlights the psychological processes inherent to development and learning, introducing the relevance of the executive functions (EFs) in explaining intelligence and giftedness. Some researchers have connected high capabilities to superior executive function development [51,52], with evidence of early and efficient use of EFs by gifted children.

Compared to their peers, these children have higher levels of cognitive flexibility, inhibitory control, working memory, and planning [53], which may be related to a stronger, flexible, and dynamic reconfiguration of brain networks in the frontoparietal regions of the executive system, promoting a superior development of fluid intelligence [54] and superior effectiveness in tasks that require working memory, flexibility, and automation of cognitive control [52,55]. Therefore, the test of the convergent validity that the Gifted Rating Scales demonstrated with measures of executive factors would be useful.

### 1.3. Aims of the Studies

In accordance with the above statements, one of the problems gifted education is facing in Greece is the challenge of accurate identification of gifted and highly talented learners via teachers’ and/or parents’ rating scales as part of a screening test. Many researchers claim that this is one of the most critical issues to be resolved before the research community of giftedness can move forward and better evaluate exceptional learners. To our knowledge, there are no empirical studies in Greece seeking to evaluate giftedness with GRS. In this vein, we conducted some studies that aim to shed light on the facets of giftedness, in kindergarteners and primary school education students in Greece, with the Gifted Rating Scales (GRS). The present paper is based on findings concerning the Preschool/Kindergarten Form (GRS-P) and consists of two studies (Study 1 and Study 2) which aimed to gain more insight into some of the psychometric properties (internal consistency reliability, structural and convergent validity) of the GRS-P form in kindergarten children in Greece.

In specific, the aims of Study 1 were: (a1) the test/confirmation of a unifactorial structure for each of the five scales of the Greek version of the GRS-P, and (a2) taking into account the finding of Benson and Kranzler (2017) [47], that a general factor (latent variable) has been found to account for most of the variance captured by the five GRS-P ratings (measured variables), another aim of this study was the examination of this possibility in a Greek sample; (b) the evaluation of the internal consistency reliability of each of the five scales of the Greek version of the GRS-P; as well as (c) the test of the convergent validity of the Greek version of the GRS-P with the Greek version of the Childhood Executive Functioning Inventory (CHEXI) [56]. In specific, the scope of the third aim of Study 1 was to evaluate the relationship between GRS-P and executive functioning working memory, and inhibition, in kindergarten children [47]. In addition, the aims of Study 2 were also: (a) the evaluation of the internal consistency reliability of each of the five scales of the Greek version of the GRS-P in a second Greek sample; and (b) the test of the convergent and discriminant validity of the Greek version of the GRS-P with some psychometric tools that measure cognitive abilities and aspects of intelligence (namely, Raven CPM, Children Category Test, Athena test, and Mini-mental State Examination).

## 2. Materials and Methods

### 2.1. Study 1

In Study 1, of the present paper, the estimations of 107 kindergarten teachers (105 women and 2 men; mean age = 43.01 years; *SD* = 8.13) were evaluated with the use of the GRS-P form. Furthermore, the mean of teachers’ years of education was 15. 86 years (*SD* = 8.7), whereas there was a differentiation in terms of educational level (Master’s Degree = 61.9% of teachers, Educational Specialization = 37.1% of teachers, Doctoral Degree = 1% of teachers). As regards the time that the teacher knew the student for whom s/he completed the GRS-P, 2 teachers knew the student from 1 to 3 months (1.9%), 12 knew the student from 4 to 6 months (11.3%), 46 knew the student from 7 to 12 months (43.4%), and 46 knew the student for more than 1 year (43.4%). As regards how well the teacher knew the student for whom s/he completed the GRS-P, 4 teachers did not know the student very well (3.8%), 53 knew the student well (50.5%), and 48 knew the student very well (45.7%). Additionally, each teacher completed the Childhood Executive Functioning Inventory (CHEXI) [55] for the evaluation of his/her student’s executive functioning. More specifically, the teacher’s task was to evaluate the dimensions of the giftedness and executive functions of 107 kindergarten students (55 girls and 52 boys) (mean age in months = 67.03, *SD* = 5.09).

### 2.2. Procedure

The two questionnaires, which were addressed to the teachers, were given in an envelope to be completed by the teacher responsible for each child. Each teacher received from the researcher: (a) the information letter concerning the objectives of the research, (b) the demographic completion form, (c) the translated Greek version of the GRS-P form, and (d) the translated Greek version of the CHEXI questionnaire [56]. All participants were examined individually. Each teacher had the opportunity to choose the place, as well as the time, to complete the questionnaire. In cases where the completion took place in the presence of the researcher, assistance or clarifications were provided, where necessary. Teachers were encouraged to respond honestly to ensure the reliability of the result. Participants were recruited from schools of different regions in Greece as the random collection of the data was conducted (a voluntary task during the attendance of an Introduction to Psychology course) by students from the Department of Early Childhood Education of the University of Ioannina in Greece under the supervision of one of the authors. Since these are considered personal data, the European Union law that has existed since 28 May 2018, was applied. According to the law, the use of sensitive personal data is allowed only due to research reasons. The study’s protocol followed the principles outlined in the Helsinki Declaration and was approved by the Scientific and Ethics Committee of the University of Ioannina (25847/01/06/2021).

### 2.3. Instruments

*The Gifted Rating Scales-Preschool/Kindergarten Form (GRS-P)* [33]. The Gifted Rating Scales include a Preschool/Kindergarten Form (GRS-P) for ages 4.0 to 6.11 and are designed to be user-friendly. The GRS-P consists of five scales (namely: Intellectual, Academic, Creativity, Artistic, and Motivation) with 12 items each (60 items). Each item in every scale is rated by a teacher on a 9-point scale divided into three ranges: 1 to 3 = below average, 4 to 6 = average, and 7 to 9 = above average. This rating system allows the teacher to determine first whether the child is below average, average, or above average for each item compared to other students the same age and then to rate the student more specifically on a 3-point scale within the range.

The following is a brief description of each of the five scales included in the GRS-P: Intellectual Ability Scale.

This scale measures the teacher’s ratings of a student’s verbal and/or nonverbal mental skills, capabilities, or intellectual competence. Academic Ability Scale: This scale measures the teacher’s ratings of the student’s skill in dealing with factual and/or school-related material. Creativity Scale: This scale measures the teacher’s ratings of the student’s ability to think, act, and/or produce unique, original, novel, or innovative thoughts or products. Artistic Talent Scale: This scale measures the teacher’s ratings of the student’s potential for, or evidence of, ability in drama, music, dance, drawing, singing, playing a musical instrument, and/or acting. Motivation Scale: This scale refers to the teacher’s perception of the student’s persistence, desire to succeed, tendency to enjoy challenging tasks, and ability to work well without encouragement. For the translation of the GRS-P inventory in Greek, by Thomaidou and Papantoniou [57], the International Test Commission (ITC) guidelines (www.intestcom.org, accessed on 9 November 2022) were followed. The back translation procedure was also followed so as to eliminate any inconsistencies that would disrupt the accuracy of the results.

*Functioning Inventory (CHEXI)* [56]. The CHEXI consists of 26 items. In this questionnaire, all the sentences/questions are associated with problems in executive functions, which can be perceived in children’s daily functions. More specifically, the 26 items of the CHEXI had initially been divided into four a priori subscales based on Barkley’s (1997) [58] hybrid model: working memory (11 items), planning (4 items), inhibition (6 items), and regulation (5 items). However, after conducting exploratory factor analyses and excluding items 25 and 26, Thorell and Nyberg (2008) [56] provided a two-factor solution of two clear factors which were easily interpreted. The first factor was interpreted as working memory (CHEXI-WM) in that planning is often considered a more advanced working memory function [58]. The second factor included the two subscales, tapping inhibition and regulation of motivation. Together, these items can be interpreted as measuring both the cognitive and motivational aspects of inhibitory control, and the second subscale was, therefore, named inhibition (CHEXI-I). In this questionnaire, teachers or parents should determine how well each sentence describes the child based on a specific 5-point Likert scale (1 = absolutely not true, 2 = not true, 3 = partially true, 4 = true, and 5 = absolutely true). The questionnaire was translated to Greek by Papantoniou and colleagues and the International Test Commission (ITC) guidelines (www.intestcom.org, accessed on 9 November 2022) were followed. A back translation procedure was also followed for the elimination of any inconsistencies that would disrupt the accuracy of the results. The factor structure of the Greek version of CHEXI was tested with CFA in a broad sample of 187 kindergarten and primary education Greek teachers, part of whom were the 107 kindergarten teachers who participated in the present study (Study 2). To evaluate the structural validity of the Greek version of the Childhood Executive Functioning Inventory, a confirmatory factor analysis was conducted for the data collected from the 24 items that constituted it after the exclusion of items 25 and 26, to verify its aforementioned two-factor structure. The implementation of the CFA was conducted in the statistical program EQS 6.1. [59] and was performed on a covariance matrix of the 24 items, using the Maximum Likelihood estimation procedure. The CFA began with the examination of the metric model, against which the two factors showed no correlations. The metric model was not acceptable according to various indices of fit. Nevertheless, all parameters were found to be statistically significant (*p* < 0.05) in this model. Then, the structural model was checked, according to which there were correlations between the two factors. The indicators in this model were improved compared to the previous model and were marginally acceptable: χ^2^(246, *N* = 187) = 427.99, *ρ* < 0.000, CFI = 0.93, SRMR = 0.06, RMSEA = 0.06 (CI 90% 0.05–0.07) [45,48]. The correlation between the factors was statistically significant. Thus, the existence of the two factors, according to the original version of the questionnaire, was confirmed. These findings are aligned with the findings of Catale et al. 2013 [60] for the French Childhood Executive Functioning Inventory, and Kompa (2014) [61] for the Greek version of the Childhood Executive Functioning Inventory. Furthermore, the reliability of the internal consistency of the Greek version of the CHEXI was evaluated with Cronbach’s alpha coefficients. The Cronbach’s *α* internal consistency for the subscale of working memory was excellent at 0.93, as well as for the subscale of inhibition at 0.90.

## 3. Results

### 3.1. Test of the GRS-P Factor Structure

As regards the sample size requirements, for SEM techniques, it is recommended as a rule of thumb that there be at least five observations per estimated parameter [62]. Given that the GRS-P consists of five scales (namely: Intellectual, Academic, Creativity, Artistic, and Motivation) with 12 items each, the sample size for confirmatory factor analysis at the item level, concluding the 60 items of the 5 scales, had to exceed 300 observations. Thus, taking into account the relatively small size of the sample, we did not conduct a total CFA concluding 60 items. Alternatively, in order to verify the one-factor structure of each of the five scales of the GRS-P for the Greek kindergarten teachers’ sample of Study 1, a set of five confirmatory factor analyses was conducted for the data collected from the 12 items that constitute each of the five GRS-P scales. The implementation of each CFA was conducted in the statistical program EQS 6.1. [46] and was performed on a covariance matrix of the 12 items of each scale, using the maximum likelihood estimation procedure. The Wald test was used to suggest more restricted models. A nonstatistical significance of the χ^2^-test indicated that the implied theoretical model significantly reproduced the sample variance–covariance relationships in the matrix. Since this test is sensitive to sample size, model fit was also evaluated by using the root mean squared error of approximation (RMSEA). The RMSEA tests how well the model would fit the population covariance matrix. A rule of thumb is that RMSEA ≤0.06 indicates a close approximate fit. The comparative fit index (CFI), which is one of the indexes assessing the relative improvement in the fit of the researcher’s model compared with a baseline model, was also used. A rule of thumb for the CFI is that values close to 0.95 or greater may indicate a reasonably good fit for the researcher’s model. In addition, the model fit was evaluated by using the standardized root mean squared residual (SRMR). The SRMR is a measure of the mean absolute correlation residual: the overall difference between the observed and the predicted correlations. Values of the SRMR less than 0.08 are generally considered favorable [45,48]. The indices in the initial models that were conducted to test the one-factor structure of each of the five scales of the GRS-P ranged from not accepted to marginally accepted: Intellectual Ability Scale: χ^2^(54, *N* = 97) = 154.38, *ρ* < 0.000, CFI = 0.94, SRMR = 0.02, RMSEA = 0.14 (CI 90% 0.11–0.16). Academic Ability Scale: χ^2^(51, *N* = 96) = 157.43, *ρ* < 0.000, CFI = 0.92, SRMR = 0.05, RMSEA = 0.15 (CI 90% 0.12–0.17). Creativity Scale: χ^2^(54, *N* = 98) = 158.63, *ρ* < 0.000, CFI = 0.94, SRMR = 0.03, RMSEA= 0.14 (CI 90% 0.11–0.17). Artistic Talent Scale: χ^2^(54, *N* = 98) = 167.65, *ρ* < 0.000, CFI = 0.93, SRMR = 0.03, RMSEA = 0.15 (CI 90% 0.12–0.17). Motivation Scale: χ^2^(54, *N* = 99) = 191.37, *ρ* < 0.000, CFI = 0.93, SRMR = 0.03, RMSEA = 0.16 (CI 90% 0.14–0.18) [44,63]. Although all parameters of the aforementioned models were found to be statistically significant (*p* < 0.05) and standardized root mean squared residual (SRMR) values were below 0.05, also indicating a good fit for the models tested, the chi-square goodness-of-fit test was statistically significant for all the initial models resulting in a rejection of the null hypothesis of good fit. In addition, the comparative fit index (CFI) values fell to the lowest boundary of the marginal range of 0.90–0.95 and were indicative of a marginally accepted model fit [63]. Finally, the root mean squared error of approximation (RMSEA) values were above 0.10, indicating a poor fit for the models tested.

For these reasons, we proceeded with the identification of the areas of the initial models that contributed the most to the misfit. A residual analysis was conducted, and the Wald test was performed. Different models were tested, and the modifications indicated by the aforementioned tests were included in the model being tested each time. The modifications improved the fit of the final models on all indices: Intellectual Ability Scale: χ^2^(51, *N* = 97) = 61.82, *ρ* < 0.143, CFI = 0.99, SRMR = 0.02, RMSEA = 0.05 (CI 90% 0.00–0.08). Academic Ability Scale: χ^2^(43, *N* = 96) = 86.66, *ρ* < 0.000, CFI = 0.97, SRMR = 0.04, RMSEA =0.10 (CI 90% 0.07–0.13). Creativity Scale: χ^2^(49, *N* = 98) = 103.13, *ρ* < 0.000, CFI = 0.97, SRMR = 0.02, RMSEA = 0.11 (CI 90% 0.08–0.13). Artistic Talent Scale: χ^2^(42, *N* = 98) = 46.03, *ρ* < 0.309, CFI = 1.00, SRMR = 0.01, RMSEA = 0.03 (CI 90% 0.00–0.08). Motivation Scale: χ^2^(46, *N* = 99) = 87.32, *ρ*< 0 0.000, CFI = 0.98, SRMR = 0.02, RMSEA = 0.10 (CI 90% 0.06–0.12) [44,63].

In the second set of the CFA, all parameters of the final models were found to be statistically significant (*p* < 0.05) and the standardized root mean squared residual (SRMR) values were below 0.05, indicating also a good fit for the models tested. In addition, the comparative fit index (CFI) values were greater than 0.95, indicating a reasonably good model fit. The statistically significant chi-square goodness-of-fit test and the nonsignificant RMSEA values that were found in three of the final models, could be interpreted in accordance with the acceptable model fit. Since *N* was somewhat small, they may be of less concern if all other indices were in a range suggesting a “good” model fit [63]. It is important to mention that the RMSEA is known to produce artificially large values for models with few degrees of freedom and small sample sizes [46]. Given that our GRS-P model had a small sample size, this statistic was likely inflated and thus not indicative of model misfit [28].

It should be noted that the aforementioned set of the CFA at the item-level data, although they fully verified the unifactorial structure proposed by Pfeiffer and Jarosewich [33] for each of the five scales of the Greek version of the GRS-P, were limited as regards the identification of the number of the GRS-P organization’s underlying factors. Since the unifactorial structure for each of the five scales of the Greek version of the GRS-P was verified at the item-level data, and taking into account the finding of Benson and Kranzler [47] at the scale-level data (not at the item-level data) that a general factor (latent variable) has been found to account for most of the variance captured by the five GRS-P ratings (measured variables), we followed Benson’s and Kranzler’s methodology [47]. In specific, we applied both an exploratory factor analysis (EFA) and confirmatory factor analysis (CFA), at the scale-level data, to all five scales of the Greek version of the GRS-P in order for us to be able to identify the number of their organization’s underlying factors (latent variables). Since both the EFA and CFA were not run at the item-level but at the total scores for the aforementioned verified factor structure of each scale of GRS-P, the intellectual ability, academic ability, creativity, artistic talent, and motivation scales were treated as measured (observed) variables first in the EFA, and then in the CFA that conducted the scale-level data.

As regards the conduction of the EFA, we estimated the sampling adequacy through the Kaiser–Meyer–Olkin test (K.M.O. = 0.914) and Bartlett’s test of sphericity (χ^2^ = 472.875, *df* = 10, *p* < 0.001), and we also used the scree plot to determine the number of factors to retain in the analysis. For the factor extraction of the factors, a principal component analysis (PCA) with orthogonal Varimax rotation was applied, due to the finding of Benson et al. (2017) [47] on the one-factor solution. The analysis yielded one factor with an eigenvalue greater than 1.00. The eigenvalue for the first factor was 4.31, and the percentage of the explained variance was 82.95%. All other eigenvalues fell well below the minimum criterion for retention. This finding is similar to Benson et al.’s [47] EFA one-factor solution. The exploratory factor analysis is presented in Table 1.

Finally, a CFA was used to confirm the aforementioned model with a single general factor. Results revealed that the one-factor model provided an excellent fit for the Greek version of GRS-P data [χ^2^(5, *N* = 71) = 1.25, *ρ* < 0.940, CFI = 1.00, SRMR = 0.01, RMSEA = 0.00 (CI 90% 0.00–0.04)]. This finding is similar to Benson et al.’s (2017) [47] CFA one-factor solution. R^2^ values for the general factor ranged from 0.66 to 0.93. Standardized loadings and residual variances for observed variables are presented in Figure 1.

### 3.2. Test of the GRS-P Internal Consistency Reliability

In the second step of statistical analyses, the internal consistency reliability of the GRS-P was evaluated with Cronbach’s alpha coefficients. The Cronbach’s internal consistency of all scales of the Greek version of the GRS-P, for the corresponding sample of Study 1, ranged between 0.97 and 0.98. The alpha internal consistency coefficient of all scales was excellent, and research findings were closely aligned with the results found by Pfeiffer et al. [33]. The alpha internal consistency coefficients of all scales are presented in Table 2.

### 3.3. Test of the GRS-P Convergent Validity

Finally, in order to test the convergent validity of the Greek version of the GRS-P with the Greek version of the Childhood Executive Functioning Inventory (CHEXI) [56], the Pearson correlation coefficient was conducted. The application of the Pearson correlation analysis revealed statistically significant moderate and negative correlations for the working memory and inhibition subscales of CHEXI with all the GRS-Ps. The CHEXI questionnaire evaluated the deficits in executive functions (working memory and inhibition impairments), and for this reason, the correlations of its subscales with the GRS-P had negative indices. Table 3 depicts the correlations between the five scales of the Greek version of the GRS-P with the working memory and inhibition subscales of CHEXI.

### 3.4. Study 2

#### 3.4.1. Participants

Study 2 included 26 kindergarten children (12 boys and 14 girls) (mean age in months = 69.04 months, *SD* = 3.94 months), who had simultaneously participated (as part of a broader sample) in another study that tested the hypothesis of retrogenesis. In this first phase of Study 2, which has also been included in the study of the theory of retrogenesis, all the young participants were tested with some psychometric tools that measured cognitive abilities and aspects of intelligence. Specifically, they completed a brief version of the Athena test for an estimate of their overall cognitive functioning to be provided. More specifically, their general cognitive ability was assessed with the subtests of language analogies, vocabulary, and pattern copying, whereas auditory and visual memory was evaluated with the subtests of digit span and memory of common sequences from the standardized Athena test for the Greek population [64]. In addition, the neuropsychological maturation of the children was evaluated with the subtest “Right-Left” perception from the same test battery. Furthermore, high-order nonverbal abilities (such as intellectual functioning and abstract thinking) were evaluated with the Children’s Category Test (CCT) [65]. For the present study, only Level-1 of the CCT was administered. Nonverbal intelligence was evaluated with Raven’s Educational Colored Progressive Matrices (CPM) [66]. All the young participants, additionally to the Educational CPM, completed the Greek version of the Mini-Mental State Examination (MMSE) [67] for a brief estimate of their overall cognitive functioning to be provided. The test was modified for the present study on children. In the second phase of Study 2, after a one-week duration, the GRS-P was completed by the teachers (3 kindergarten teachers) of the 26 kindergarten participants based on their direct observations.

#### 3.4.2. Procedure

The children, who were recruited for Study 2, came from the broad area of Epirus in Greece, and they were attending a private school, which was indicative of high educational and socioeconomic background. Prior to the study and in collaboration with the school committee, parents gave their written statement of consent for the participation of their children in this study, and then they completed an individual–demographics form. The evaluation consisted of two distinct phases. All the participants were individually examined in the presence of a trained experimenter in a quiet room in the school. In the first phase, their verbal and nonverbal cognitive abilities were evaluated. To avoid the mental fatigue of the participants, a general cognitive ability evaluation was conducted in two distinct individual trials. In the second phase, one week after the children’s general cognitive assessment, all teachers of the classes in which the participants studied were requested to complete the CRS-P for each of the participating students. All children were attending regular classrooms, without a history of learning difficulties. Teachers individually completed the ratings based on their observations and not on their inferences. No time limit was assigned for the completion of the scales, and all participants were informed that they were free to withdraw from the evaluation process at any time. Since these are considered personal data, the European Union law that has existed since May 28, 2018, was applied. According to the law, the use of sensitive personal data is allowed only due to research reasons. The study’s protocol followed the principles outlined in the Helsinki Declaration and was approved by the Scientific and Ethics Committee of the University of Ioannina (25847/01/06/2021).

#### 3.4.3. Instruments

*The Gifted Rating Scales-Preschool/Kindergarten Form (GRS-P)* [33]. The GRS-P form has been presented in Study 1 of the present paper.

*Athena test (AT)* [68]. The Athena test (AT) is a Greek diagnostic tool for the evaluation of cognitive abilities of children aged from 5 to 9 years. The AT is a multidisciplinary test, which provides the ability to describe in detail the current cognitive level of the examinee in key areas of development and identifies specific areas which are deficient. The subtests and the philosophy of this specific battery are closely aligned with WISC-III, and the AT has been standardized in the Greek population [68]. The AT is divided into fifteen (15) subtests, fourteen (14) core, and one (1) supplemental, which assess a wide range of abilities in motor, perceptual, cognitive, and psycholinguistic domains. The inclusion of the six subtests described here was made on the basis that they were complementary to the other measures used in the study, namely the Raven’s Educational CPM, the Children’s Category Test, and the Mini-Mental State Examination. Furthermore, the selective administration of scales of the AT was provided as a possibility by its manufacturers [67] in cases where only specific scales were aligned with the research purposes.

The six AT subtests that were used in the present study are the following:*Language Analogies*. Language analogies consist of 32 pairs of sentences. Each pair contains four meanings that relate to each other (e.g., shape) and form an analogy expressed in words. For each pair of sentences, the first sentence is complete while the second is incomplete. The participant’s task is to find the missing word in this second sentence. Every correctly answered item results in 1 point. It is a test of verbal concept formation, which reflects a child’s abstract verbal classification abilities.*Pattern Copying*. The participant is presented with three geometrical patterns that vary in difficulty. The correct pattern copying requires increasing visuo-conceptual maturity with every pattern. The drawings are scored by three morphological characteristics: (a) general resemblance to standard, (b) shape orientation, and (c) equality of different parts. Each drawing is scored by 0–3 points.*Vocabulary*. It consists of 20 items: words, both concrete and abstract, for which the subject needs to give a definition. They vary in difficulty from the simplest to more difficult words. Each answer is scored by 0–2 points depending on the accuracy of the definition. This test assesses word knowledge, language development, and long-term memory.*Digit Span*. It includes 16 items: sequences of numbers that gradually get longer. Participants are given the sequences orally and are asked to repeat them as heard. Correctly repeated items are given a score of 2 or 1, at the first or second attempt, respectively. This test assesses short-term phonological memory.*Memory of Common Sequences*. This is a supplemental subtest in the AT test. The child is asked to name parts of sequences encountered in everyday life (days of the week, months of the year, etc.). Every correctly answered item results in 1 point.*“Right—Left” Perception* (neuropsychological maturation). Two items in the form of instructions are to be executed by the participants, indicating whether they can distinguish and orientate accordingly the right and left “side” of the body, either their own (direct perception) or the examiner’s (“mirror” perception). Every correctly answered item results in 1 point.

*Children’s Category Test (CCT-1)* [65]. The CCT is an individualized administered test designed to assess deficits in nonverbal learning, memory, concept formation, and problem-solving abilities with novel material. It provides information on the child’s ability to develop alternative solutions; change problem-solving strategies; benefit from the experience; and develop, test, and modify hypotheses. This constellation of cognitive processes is highly related to fluid intelligence. The CCT may be used to determine whether a child is able to perform the aforementioned processes despite the existence of learning disorders, verbal or motor deficits, neurological deficits, or emotional handicaps. The Children’s Category Test consists of two levels. Level one (CCT-1) was developed for children of 5–8 years of age, and Level two (CCT-2) was intended for ages 9–16. Both levels require the child to create and modify strategies for responding to visual stimuli based on corrective feedback. The CCT-1 (Level 1), which was administered to participants of Study 2, is presented in booklet form, and consists of 80 items distributed across 5 subtests. The test requires the child to identify the conceptual rule that underlies each subtest and apply that concept to answer each item correctly. The child is shown a series of pictures that are intended to suggest a particular color, and they must respond by pointing to (or verbally identifying) one of four colors printed on a response card. The child receives immediate feedback as to the correctness of each response, and it is expected to use this feedback to determine the conceptual rule underlying the subtest. Subtest I can be employed as a practice test to determine if the child understands the task and is able to provide appropriate responses, while subtest V requires the child to remember and apply the principles of the previous subtests. The conceptual rules underlying the subtests include the following: color recognition (subtest I), determination of the relative quantity of a specific color (subtest II), identification of the oddity in shape or size (subtest III), identification of the missing color (subtest IV), and review of the principles presented in subtests I through IV (subtest V).

*Raven’s Educational Colored Progressive Matrices (R-CPM*) [69]. Raven’s Educational CPM/CVS consists of two subtests: the Colored Progressive Matrices (CPM) subtest, which is used for measuring nonverbal intelligence, and the Crichton Vocabulary Scales (CVS) subtest, which measures verbal intelligence. The test is standardized in the Greek population. In the present study, only the Colored Progressive Matrices subtest was used. It is considered an appropriate instrument for measuring the nonverbal intelligence of young children ranging in age from 4 to 11 years. The book form of the Educational CPM contains three sets (A, AB, and B) of 12 items of colored large-print drawings each. In each item, participants are presented with an incomplete design and six alternatives, among which one must be chosen that best completes the design. Each correct answer scores one (1) point. Therefore, each participant could collect 12 points in each subscale, that is, 36 points in total (score range 0–36).

*Mini-Mental State Examination (MMSE)* [70]. The MMSE is a 30-point scale assessing orientation to time and place, registration, attention and calculation, recall, language, and visuo-construction. Administration time typically is approximately 10 min. The MMSE has been translated into Greek and standardized in the Greek population [70]. A version of the mini-mental state examination was adapted for children according to Savvidou et al., (2016) [71]. The modified version of the MMSE test was part of the tools that were also administered to the young participants of Study 2 for the simultaneous data gathering for the aforementioned parallel research project which tested the retrogenesis theory. The MMSE for preschoolers assesses spatial and temporal orientation, verbal, and visual memory, reading prerequisites, numerical knowledge, praxis, body representation, and executive functions through 16 items in a short period of time.

## 4. Results

### 4.1. Test of the GRS-P Internal Consistency Reliability

In the first step of statistical analyses, the internal consistency reliability of the five scales of the Greek version of the GRS-P was evaluated with Cronbach’s alpha coefficient. As shown in Table 4, Cronbach’s internal consistency of all scales of the GRS-P, for the corresponding sample of Study 2, ranged between 0.98 and 0.99. The alpha internal consistency coefficient of all scales was excellent, and the research findings were closely aligned with the results found by Pfeiffer et al. [33].

### 4.2. Test of the GRS-P Convergent Validity

To examine the convergent and discriminant validity of each of the five GRS-P scales with the colored progressive matrices (CPM), the Athena test (AT), the Children’s Category Test (CCT-1), and the Mini-Mental State Examination (MMSE), the Pearson correlation coefficient between them were calculated. The variables were created based on the sum of the answers to each of the six subtests of the AT, to each of the CCT-1 five subtests, and to the total CCT-1, as well as to the total CPM and MMSE tests. The application of correlation analysis revealed a lack of significant correlations between Colored Progressive Matrices (CPM) and all the subscales of GRS-P. A full correlation matrix among measures is presented in Table 5.

The application of the Pearson analysis revealed moderate positive statistically significant correlations between only the subtest of the digit span of the AT and three of the scales of GRS-P (Intellectual ability *r* = 0.47, Academic ability *r* = 0.43, Creativity *r* = 0.46). A full correlation matrix among measures is presented in Table 6.

Additionally, the application of the Pearson analysis revealed a moderate positive statistically significant correlation between the MMSE test and only the Intellectual ability scale (*r* = 0.43). A full correlation matrix among measures is presented in Table 7.

On the contrary, moderate negative statistically significant correlations emerged between the scales of GRS-P (except for the Creativity scale) and the total sum of the CCT-1 test. In addition, negative statistically significant correlations were found between the fourth subtest of CCT-1 (subtest that estimates deficits in problem-solving abilities) and all the scales of GRS-P (Intellectual ability *r* = −0.534, Academic ability *r* = −0.459, Creativity *r* = −0.401, Artistic talent *r* = −0.470, Motivation *r* = −0.540). A full correlation matrix among measures is presented in Table 8.

To further evaluate the relationship between the GRS-P form and CPM, CCT, MMSE, and the subscales of the AT, a path analysis was performed. Because of the relatively small sample size of the group, the covariance matrix was based on total scores (measured variables), namely, total raw scores for CPM, CCT, MMSE, AT subscales’ tests, and the five scales of the GRS-P form. A path analysis was conducted in EQS Version 6.1 [46] using the maximum likelihood estimation procedure and performed on the covariance matrices, which stemmed from the total sample and the group of participants. In the equations of the path analysis, the performances of the children in the different tests were defined as independent variables (exogenous variable), and the ratings of the teachers in the five scales of GRS-P as dependent variables (endogenous variable). The direction of the relationship began with the performance in different cognitive tests towards the variables of the five scales of GRS-P. Different path analysis models were calculated, in which the findings of the three modified methods (largest standardized residuals, Lagrange test, and Wald test) of the model were integrated, each time. Due to the small number of participants, the different models calculated included only five variables at a time. Therefore, to include and control all the variables under consideration, two final models emerged. The goodness-of-fit indexes which were provided from the application of the path analysis for the first model were excellent and met the corresponding criteria χ^2^(1, *Ν* = 26) = 0.001, *p* = 0.972, χ^2^/*df* = 0.001, CFI = 1.000, SRMR = 0.003, RMSEA = 0.000 [45,48]. All the parameters’ loadings of the first final model were statistically significant (*p* < 0.05). Accordingly, the goodness-of-fit indexes which were provided for the second model were very satisfactory and met the corresponding criteria χ^2^(2, *Ν* = 26) = 0.296, *p* = 0.862, χ^2^/*df* = 0.148, CFI = 1.000, SRMR = 0.044, RMSEA = 0.000 [45,48]. All the parameters’ loadings were statistically significant (*p* < 0.05). In the first model, negative relations emerged for the performance on the children’s category test for deficits of high-order nonverbal abilities and three scales of GRS-P (Intellectual −0.449, Academic −0.395, Creativity −0.379), whereas positive relations were revealed for the digit span test from the AT and three scales of GRS-P (Intellectual 0.470, Academic 0.434, Creativity 0.461). Figure 2a depicts the relations between the children’s category test and digit span with the scales of GRS-P. Accordingly, in the second model, negative relations emerged also for the CCT performance and the other two scales of GRS-P (Creativity −0.417 and Motivation −0.445) whereas a negative relation was found for the digit span test and the scale motivation of the GRS-P (Motivation 0.192). Figure 2b depicts the relations between the children’s category test and digit span with the two subscales of GRS-P.

## 5. Discussion

The aims of Study 1 and Study 2 of the present paper were the examination of the internal consistency reliability, the factorial validity, and the convergent and discriminant validity of the Greek version of the Preschool/Kindergarten Form (GRS-P) of Gifted Rating Scales (GRS): an assessment tool strongly aligned with the identification of high-achieving or gifted children. Results of the two studies revealed that the very high internal consistency of the scales as well as their good factorial and discriminant validity in relation to the general mental ability of children emphasize the fact that the GRS-P is a reliable and valid tool for assessing gifted learners by their teachers in the Greek cultural context. More specifically, the indices of the internal consistency reliability for the five scales of the Greek version of the GRS-P, for both samples of the present paper, were excellent and ranged between 0.96 and 0.99. The above indices are extremely high and could suggest redundancy in the items. However, it should be noted that they are comparable to those found in the United States by Pfeiffer et al. [33], as well as by Karadag and Pfeiffer [41] for the Turkish adaptation, and by Siu [42] for the Chinese adaptation. The purpose of Study 1 also was: (a1) the test/confirmation of a unifactorial structure for each of the five scales of the Greek version of the GRS-P, and (a2) taking into account the finding of Benson et al. [28], that a general factor (latent variable) has been found to account for most of the variance captured by the five GRS-P ratings (measured variables). Another purpose of this study was the examination of this possibility in a Greek sample. The CFA applied separately (due to the small sample size) to each of the five GRS-Ps also revealed that, as in the original version, the Greek version of the GRS-P, which is addressed to teachers, assesses giftedness in children, as they are structured in five separate scales/factors: intellectual ability, academic ability, creativity, artistic talent, and motivation. Based on the above results, the GRS-Ps were found to retain the factorial structure proposed by their manufacturers, when administered to the sample of the present study. This is an encouraging indication of the future normative data and uses in the Greek population as a screening evaluation tool for identifying distinct constructs of giftedness in young children.

However, it should also be taken into account that, as regards the test of aim a2, the application of EFA and CFA revealed the presence of a large general factor, which was found to explain a large proportion (82.95%) of the GRS-P variance. This finding is in accordance with Benson et al.’s [47] one-factor solution, which is indicative of the need for a total GRS-P score to be developed as well. Our findings may suggest that the GRS-P primarily reflects a general cognitive ability with a multidimensional conceptualization of giftedness [72]. The multidimensional profile of giftedness could be explained by the fact that teachers engage in the rating of a number of student characteristics, such as social-emotional skills [71], motor skills [72], and artistic abilities [73]. Consequently, a teacher bases his or her judgment of both these constructs on a certain global perception of students’ abilities [74]. At the same time, it turns out that it affects the teacher’s judgment even in other unrelated areas, such as artistic abilities, creativity, and persistence. Additionally, a teacher associates his or her judgment with certain key characteristics. Such cues may include, e.g., the student’s previous academic results, his or her social competencies, working habits, socioeconomic status, etc. In his or her assessment, the teacher then ascribes a certain weight to each of these cues, which affects his or her resulting judgment. We assume that whether the teacher judges cognitive abilities or academic achievement, he or she comes from a place of a global impression of the student, which is primarily based on readily observable cues such as academic performance and behavior within the academic environment. Other cues may also include the student’s cognitive abilities and creativity (intelligence in general). As a result, a child’s real academic achievement reflects his or her cognitive abilities (fluid reasoning, working memory, learning efficiency, etc.), other noncognitive influences (motivation, social support, etc.), academic knowledge, and nonintellectual abilities [75,76]. Consequently, the above interpretations are aligned with the tripartite model of giftedness, according to which giftedness can be viewed through the lens of high intelligence, outstanding accomplishments which are characterized as nonintellectual factors (creativity, academic passion, persistence, motivation), and through the lens of the potential to excel [5].

As regards the aim of Study 1 to test the convergent validity of the Greek version of the Preschool/Kindergarten Form with the Childhood Executive Functioning Inventory, the negative correlations, that were found between the five GRS-P scales and the two CHEXI subscales strongly support the claim that teachers can accurately identify that working memory and inhibition are closely aligned with the profile of a gifted child. In addition, the present findings are consistent with Kornmann et al. [77], which confirmed the importance of working memory for the characterization of children as highly talented by their teachers. This finding supports the idea that working memory measurements can be a prognostic factor for children’s future academic achievement and IQ performance [59,76,77]. On the other hand, there is a research consensus that highly talented or gifted children register high performances in working memory measurements in comparison with mainstream children [78]. As concerns inhibition, the literature review reveals that gifted students score higher on mental attention tasks than their mainstream peers and have better performance on active mental-attentional suppression of task-irrelevant information [66]. This finding strongly supports the data of the current research. Furthermore, the above findings underlie that there is a strong connection between giftedness and executive functioning, and that teachers can identify that a gifted student can demonstrate their extraordinary abilities as early as preschool age, irrespective of cultural background and socioeconomic status. In conclusion, the research points out that working memory should be considered a key factor in the field of giftedness [79].

As regards the aim of Study 2 to test the convergent validity of the Greek version of the Preschool/Kindergarten Form with the cognitive measures of the CPM, CCT, AT, and MMSE, the scales of Intellectual, Academic, and Creativity of GRS-P were found to correlate with a moderate positive relationship with the subtest of digit span of AT test, and the scale Intellectual was found to be related to the MMSE, as well. The findings revealed that the teachers’ ratings for giftedness are based on their intellectual performances. This is a research finding that is closely aligned with many studies by Pfeiffer et al. [33], according to which the scores of the GRS-P are correlated with the scores of several recently revised intelligence and performance scores. The above findings support previous research findings indicating that the examination of the correlations between the scores of the two GRS and the scores of the total IQ indices, as well as the individual scores on the WPPSI-III, WISC-IV, and WIAT-II subtests, reveals the existence of high internal validity [33]. The existence of this similar pattern is likely due to the high internal correlation observed between the scales of Intellectual and Academic ability [33]. Additionally, the findings support the idea that teachers in the first line are able to recognize that children with low performance on cognitive measures cannot be considered gifted or high-talented learners.

Additionally, a path analysis revealed that the performance of preschool children in the Children’s Category Test (CCT-1) interprets 14.5% to 20.5% of the variation in teachers’ ratings, at all subscales of the GRS-P. These negative relations mirror an alignment between teachers’ ratings for giftedness and difficulties/deficits in cognitive measures evaluating fluid intelligence for kindergarten students. The Children’s Category Test is used to evaluate learning, memory, and problem-solving skills to measure aspects of fluid intelligence [57]. More specifically, the ratings of teachers for gifted preschool children are affected by their performance on a variety of different cognitive measures such as CCT-1, and MMSE. The pattern of this relationship shows that the low performance by the preschool students concerning nonverbal learning, concept formation, and problem-solving constructs a nongifted profile according to teachers’ ratings. Thus, teachers can recognize that gifted students can demonstrate their extraordinary abilities as early as preschool, irrespective of their cultural background and socioeconomic status [28]. Furthermore, the consensus on interpreting difficulties in nonverbal abilities and neuropsychological maturation by the teachers can be explained by the fact that teachers emphasize academic and intellectual performances, resulting in a “synchronization” among the cognitive abilities of children and teachers’ ratings for giftedness.

On the other hand, the performance of the kindergarten students in the Athena Test and more specifically in the Digit Span interprets 3.5% to 22% of the variation of the teachers’ ratings, in four of the five scales of GRS-P. These positive relations indicate the general connection between teachers’ ratings and the short-term phonological memory of preschool children. The literature review reveals that young children’s ability to retain auditory information in their short-term memory is closely related to the richness of their vocabulary [67,68]. Measurements of short-term phonological memory at a given time can predict the subsequent performance of young children in vocabulary and word-learning projects [80].

The present paper is based on findings concerning the Preschool/Kindergarten Form (GRS-P) and consists of two studies (Study 1 and Study 2) that attempt to go one step further by evaluating teachers’ rating scales as part of the screening test regarding high-level potential students in Greece and to gain more insight into some of the psychometric properties (internal consistency reliability, structural and convergent validity) of the Greek version of the GRS-P form. There are a growing number of researchers that support the idea that teachers can play an important role in the process of highlighting gifted students, as they are the ones involved in their daily educational process. Regular class teachers are the school’s first line in the identification process for these high-achieving students. Of foremost importance is the fact that teachers, who know their students well, can identify students who do not perform well on cognitive tests. Furthermore, teachers are considered very important in the process of highlighting gifted students [81].

Under the aegis of giftedness, the investigation of the psychometric properties of the Greek version of the GRS-P could help to form a validity scale for the identification of privileged and talented learners, as a new screening approach in parallel with the evaluation of cognitive abilities. Although a plethora of diagnostic assessment tools has been developed, from time to time, for the identification of highly talented learners, the need for the development of a screening tool with sufficient reliability and validity remains of vital importance [14,15,82].

As a result, the potential of these well-designed scales could ensure the accurate identification of talented learners in the Greek school environment, which cannot be detected, at the screening level, through other cognitive tests, such as intelligence or academic assessment tools [83,84]. In this vein, it is fundamental to mention that, with GRS-P, teachers can be aware of the gifted abilities of their students, and their ratings could be reliable information for implementing a variety of strategies embedded in the educational curriculum for highly talented learners [85,86]. Thus, the GRS-P seems to be a valuable rating scale that assists in determining eligibility for gifted programs. Additionally, it increases the identification process by providing multiple sources of data concerning giftedness, as teachers’ ratings are of vital importance because their extraordinary abilities sometimes cannot be evaluated with typical IQ measurements [47]. Finally, the data from the GRS-P, simultaneously with the evaluation of intelligence, can help teachers to plan and implement essential strategies for gifted and talented students embedded in the educational curriculum. Additionally, teachers have more experience with many children and, therefore, have an ambiguous database of essential information through which they can estimate different types of problem behaviors. In terms of executive function, teachers, more often than parents, can observe children in situations that require a high control of executive functions.

Of foremost interest, according to the abovementioned statements, is the fact that the evaluation process of identifying highly talented children should be an amalgam of an assessment, which includes children’s cognitive ability and school performance tests, as well as scales for assessing their giftedness by their teachers. Preschool gifted children do not necessarily have as many characteristics as older gifted children because they have had fewer opportunities to learn and achieve academic achievement [51]. As a result, school performance tests are rather inappropriate for use in identifying young, gifted children. In other words, to illuminate giftedness, we recommend standardized intelligence tests (cognitive abilities), as well as the assessment of giftedness by teachers with the GRS, which is also suitable for use with young children. From an educational point of view, future research should aim to develop learning environments that stimulate active learning and at the same time require a high level of working memory or executive control, respectively, to provide support for the learning performance of gifted students.

Some limitations of the two studies that are presented in this paper are the small sample of children who participated in both studies, and the lack of creativity, artistic abilities, and motivation measures that could also be included to enhance the test of discriminant validity. As these preliminary studies are part of a broader experimental design concerning the GRS in the Greek population, further research data are under evaluation. Although more research is needed to further validate and refine the Greek version of the GRS-P and to replicate our current findings, the results of our studies, with the size of the samples used and the breadth of the variables examined [47], show that the GRS-P is a useful instrument for measuring giftedness in the Greek cultural context and could provide an initial base in examining cross-cultural differences in aspects of giftedness, as well as in extending evolving cross-cultural research endeavors on Pfeiffer et al.’s [33] theoretical approach underlying the Gifted Rating Scales [47].

## Figures and Tables

**Figure 1 diagnostics-12-02809-f001:**
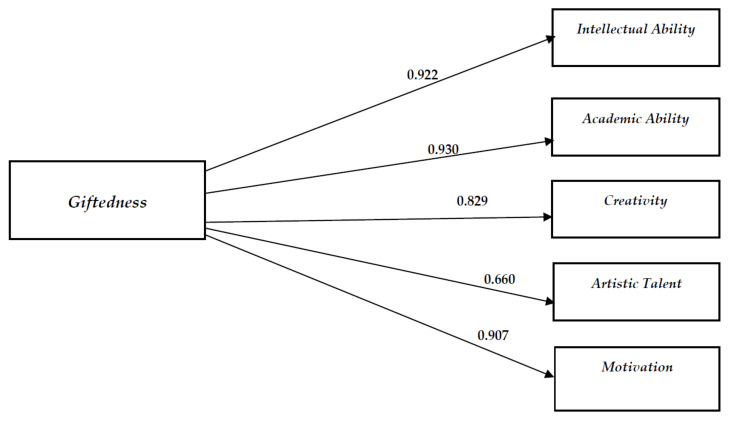
Structural model and standardized loadings for the GRS-P form.

**Figure 2 diagnostics-12-02809-f002:**
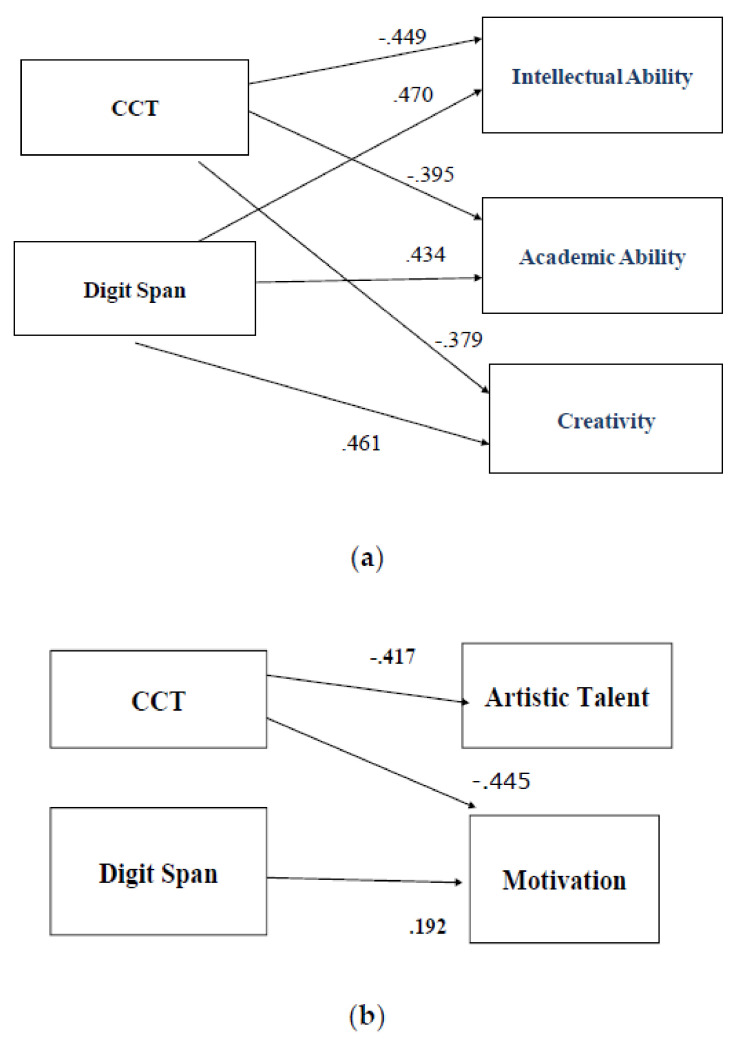
(**a**) The relations of the children’s category test and the digit span subtest to the intellectual, academic, and creativity GRS-P scales. (**b**) The relations of the children’s category test and the digit span subtest to the artistic talent and motivation GRS-P.

**Table 1 diagnostics-12-02809-t001:** The exploratory factor analysis of the GRS-P.

GRS-P	Factor
Intellectual Ability	0.957
Academic Ability	0.944
Creativity	0.893
Artistic Talent	0.801
Motivation	0.950

**Table 2 diagnostics-12-02809-t002:** Internal consistency reliability coefficients of the GRS-P.

GRS-P	Cronbach’s α
Intellectual Ability	0.980
Academic Ability	0.965
Creativity	0.980
Artistic Talent	0.979
Motivation	0.983

**Table 3 diagnostics-12-02809-t003:** Correlations between GRS-Ps and working memory and inhibition subscales of Childhood Executive Functioning Inventory (CHEXI).

	Intellectual Ability	Academic Ability	Creativity	Artistic Talent	Motivation
CHEXI Working Memory	−0.738 **	−0.739 **	−0.596 **	−0.573 **	−0.736 **
CHEXI Inhibition	−0.566 **	−0.583 **	−0.438 **	−0.484 **	−0.586 **

** Correlation is significant at the 0.01 level (2-tailed).

**Table 4 diagnostics-12-02809-t004:** Internal consistency reliability coefficients for GRS-P.

GRS-P	Cronbach’s α
Intellectual Ability	0.99
Academic Ability	0.98
Creativity	0.99
Artistic Talent	0.99
Motivation	0.99

**Table 5 diagnostics-12-02809-t005:** Correlations between GRS-P and colored progressive matrices test (CPM).

GRS-P	Colored Progressive Matrices (CPM)
Intellectual Ability	0.245
Academic Ability	0.295
Creativity	0.308
Artistic Talent	0.120
Motivation	0.131

Correlation is significant at the 0.01 level (2-tailed). None of the presented correlations is significant.

**Table 6 diagnostics-12-02809-t006:** Correlations between GRS-P and Athena test (AT).

	Intellectual Ability	Academic Ability	Creativity	Artistic Talent	Motivation
Language Analogies	0.310	0.287	0.360	0.092	0.143
Pattern Copying	0.190	0.249	0.211	0.317	0.200
Vocabulary	0.297	0.223	0.374	0.133	0.149
Digit Span	0.472 *	0.437 *	0.463 *	0.101	0.276
Memory of Common Sequences	0.078	0.137	−0.068	0.014	0.076
Right-Left Perception	0.158	0.198	0.033	0.291	0.355

* Correlation is significant at the 0.05 level (2-tailed).

**Table 7 diagnostics-12-02809-t007:** Correlations between GRS-P and MMSE.

GRS-P	MMSE
Intellectual Ability	0.431 *
Academic Ability	0.386
Creativity	0.327
Artistic Talent	0.273
Motivation	0.346

* Correlation is significant at the 0.05 level (2-tailed).

**Table 8 diagnostics-12-02809-t008:** Correlations between GRS-P and CCT-1.

	Intellectual Ability	Academic Ability	Creativity	Artistic Talent	Motivation
Intellectual Ability					
Academic Ability	0.954 **				
Creativity	0.920 **	0.901 **			
Artistic Talent	0.793 **	0.835 **	0.811 **		
Motivation	0.848 **	0.919 **	0.822 **	0.907 **	
CCTest 1st Subtest	0.121	0.120	0.156	0.096	0.070
CCT 2nd Substest	−0.005	−0.003	0.015	−0.062	−0.007
CCT 3rd Subtest	−0.219	−0.222	−0.322	−0.243	−0.203
CCT 4th Subtest	−534 **	−0.459 *	−0.401 *	−0.470 *	−0.540 **
CCT 5th Subtest	−0.375	−0.325	−0.324	−0.244	−0.272
CCT Total	−0.452 *	−0.397 *	−0.382	−0.417 *	−0.439 *

** Correlation is significant at the 0.01 level (2-tailed). * Correlation is significant at the 0.05 level (2-tailed).

## Data Availability

Not applicable.

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
