# Peer review of "The Gifted Rating Scales-Preschool/Kindergarten Form (GRS-P): A Preliminary Examination of Their Psychometric Properties in Two Greek Samples"

_diagnostics, 2022, doi:10.3390/diagnostics12112809_

Round 1

Reviewer 1 Report

Thanks for this interesting article. Although the researchers used several instruments, however, the samples are really small, which represents a challenge in generating trusted psychometric properties for GRS-P in the Greek culture. The manuscript needs a major revision. 

No need to write two paragraphs about the importance of giftedness.

P. 2, line 63: remove the possessive apostrophe from the "S" in "students

P. 2, line 64: What do the researchers mean by "the identification of students’ performance". It needs more clarification.

- The second paragraph in p.2 needs rewriting. The authors should consider and consult Handbook of giftedness in children (2008 and 2008). It should have a good background about assessment of giftedness.

- The researchers should be careful in this statement "The typical IQ 71tests as screening instruments for highly talented children". IQ testing is one of the established approaches to identify giftedness, though it had been criticized for decades, but it is still one of the indicators. I can't say it's a screening assessment unless it's nonverbal. So, I suggest to rewrite this section with being specific about types of IQ assessments used in identification.

- Why did the third paragraph start with working memory and then rating scales? There should be a meaningful connection here.

- The authors might benefit from the results of this study when they talk about the factor structure of the GRS-S in other cultures https://doi.org/10.1177/0162353220933005

- The researchers need to provide literature support in the introduction in relation to the relationship between executive functioning ratings and giftedness.

- In figure 1, the "g" factor might conflict with general factor for intelligence. I recommend choosing another label/name.

- The sample size in Study 1 and Study 2 is an issue here. There should be caution in generalizing the results.

- The researchers have to provide compelling literature that connects all the measurements/assessment tools they are using in the study.

Author Response

Response to Reviewer 1

Comments

Dear Reviewer thank you very much for all of your comments. You helped us to clarify the content of our work. For this reason we modify the Introduction according to the suggested reference.

Point 1: What do the researchers mean by "the identification of students’ performance". It needs more clarification.

- The second paragraph on p.2 needs rewriting. The authors should consider and consult the Handbook of giftedness in children (2008 and 2008). It should have a good background in the assessment of giftedness.

Answer to Comment 1: Thank you very much for your comments. In an attempt to clarify the importance of the identification of giftedness, as well as to shed light on the IQ and the relation of giftedness we follow your advice to use part of the theoretical background from the Handbook of Giftedness (2008). As a result, the Introduction part, (part of the second and part of the third paragraphs) was rewritten according to the content of the book. You can track all the changes as their highlighted in yellow.

Point 2:  Why did the third paragraph start with working memory and then rating scales? There should be a meaningful connection here.

Answer to Comment 2: Thank you very much for your comment. We added information about the relationship of working memory and executive function for gifted children. For example, according to Duggan and Garcia-Barrera (2015) executive function and intelligence are constructs that consist of common conceptual elements. The proper functioning of the EF is associated with high IQ. Problem-solving and insight concepts related to EF correspond to the behaviors of gifted individuals. According to the findings of Duff, Schoenberg, Scott, Russell & Adams [20] there is a strong relation between executive functioning and working memory capacity as measured by standardized neuropsychological tests. Their analyses indicated that the two cognitive domains shared more than 50% of variance, revealing that intellectual functioning is strongly related to both executive functioning and memory [20, 21, 22, 23]. Furthermore, in their empirical review, Foley et al., (2011) [24] concluded that it would be important to include measures of executive function in assessments, with the consideration that gifted students may not show clinical levels on these measures to the same degree as the general population. A review of the literature suggests that benchmarks for executive functioning could facilitate a better understanding of gifted populations.

Point 3: The authors might benefit from the results of this study when they talk about the factor structure of the GRS-S in other cultures https://doi.org/10.1177/0162353220933005

Answer to Comment 3: Thank you very much for your help. This is an article that we didn’t know. For this reason information from this specific article is incorporated in the content of our manuscript.

Point 4: The researchers need to provide literature support in the introduction in relation to the relationship between executive functioning ratings and giftedness. We have incorporated studies emphasizing on the relationship of working memory and executive function and giftedness. (see also the answer for comment 2)

Answer to Comment 4: Thank you very much for your comment. We have incorporated studies emphasizing on the relationship of working memory and executive function and giftedness.

Point 5: In figure 1, the "g" factor might conflict with general factor for intelligence. I recommend choosing another label/name.

Answer to Comment 5: Thank you for your comment. We have changed the name to Giftedness. This a latent variable reflecting common variance in ratings.

Reviewer 2 Report

The paper presents a validation of the Greek version of the GRS-P, an assessment tool meant to screening giftedness in children and filled by teachers. The main result seem to be that the GRS-P is moderately coherent with definitions of giftedness based on intelligence, working memory or inhibition capacities.

General comments

The topic is interesting both for research and practice – detecting the gifted children in school being of educational utmost importance. The study is not outstanding, but it definitely has some merits.

However, several important aspects should be addressed:

1.     First of all, I wonder why submitting such an article to Diagnostics. The paper has nothing to say on developmental disorder, any diagnostics or even medicine. It is a psychology, or educational paper, presenting a tool that could be useful for educators, psychologists, but not in psychiatry.

2.     Not being native myself, I cannot assess the language, but I found some sentences odd and got the feeling that the paper should be edited. I hope a native reviewer will be able to invalidate or confirm my impression.

3.     I found the theoretical framework unclear. After reading the Introduction carefully, the reader still has no idea about what exactly is the diversity of giftedness conceptions stated by the authors, or what definition the authors embrace. For this reason, many statement are hard to understand. For instance, it is clear that the authors consider the definition of giftedness by high intelligence (g) only as obsolete. However, they still use intelligence to argue about the (convergent) validity of the GRS-P scale. The theoretical framework should be clarified.

4.     References are sometimes irrelevant: papers on specific topics are cited as examples of general conceptual framework, where one would expect a literature review or a theoretical paper.

5.     I’m not familiar enough with all the statistics methods used in this paper to be positive. However, I found that the statistical procedures for study 1 are at least suspicious. For instance, running an EFA after a CFA on the same data sounds odd. Moreover, if my understanding is correct, the scale was revised between the CFA and the EFA, which looks like an unconventional procedure.

Specific comments

-       In the abstract, the last sentence (lines 32-36) doesn’t seem correct. This probably should be 2 sentences, but I don’t see what the “emphasized” refers to.

-       I think section 1.1. should be re-written. Some parts are not informative and could be removed (such as the sentence on line 42-43, that brings nothing either in term of information or style). The theoretical discussion is important, but remain allusive and fuzzy. More specifically, here are some points worth addressing:

o   I think the authors should read some of David Dai’s chapter. David Dai precisely synthetizes the history and the different perspectives on giftedness.

o   I wouldn’t say that the literature now shows a consensus on the idea that giftedness is multidimensional. Even now, most research are based on the IQ-definition of giftedness. So, the claim that is made on lines 58-59 should either be corrected or documented with references.

o   The categorization in two framework one being the performance and the other “alternative assessment methods” is unclear, to say the least (lines 64-65). Moreover, the given references don’t seem to be directly related to this opposition. For instance, the second one, supposed to illustrate alternative assessment methods, is a paper discussing creativity, not giftedness.

o   While reading this section the reader needs to know (1) what definition of giftedness and which framework the authors endorse and (2) is the distinction between those using IQ tests and those relying mostly on parents and teachers’ questionnaires merely methodological, or does it correspond to a deep theoretical opposition?

o   It would be nice to have a more detailed description of the multidimensional model underlying the GRS (or the Munich Model)

-       Line 120-121, the Cronbach’s alpha mentioned are extremely high, one could even say too high. This suggests redundancy in the items. This should at least be mentioned.

-       Lines 146-. If one can validate the GRS by the fact it correlates to IQ measures (to say it roughly), doesn’t it invalidate the idea that the framework or definitions of giftedness are distinct? This should be clarified.

-       Lines 164-182. Once again, the text should be clarified. The authors first state that there is no clear definition of giftedness, but still make claims on “these children” without details about the underlying definition (in this case, given that Vaivre-Douret is cited to, it should be children with high IQ).

-       Lines 183-189. Here again, the theoretical framework is elusive. What do the authors mean by “gifted children” when then call for a better identification? If gifted children are those with IQ above 130, the identification is easy. I understand this is not a definition they endorse, but they are never explicit about they own framework.

-       Line 201, I would recommend deleting “another aim of this study was the examination of this possibility in 201 a Greek sample”, since it doesn’t convey any important information.

-       Line 215. “In Study 1, of the present paper, the estimations…”: I think there should be no comma after “Study 1”.

-       Lines 214-230. I think that a table would be better to present the data on teacher-pupil relations

-       Lines 250-. The instruments are nicely and clearly described.

-       Lines 317-. I’m not that familiar with CFA, but I found it strange to run 5 different CFA on the data, to ensure that each factor is indeed a factor, but not allowing the data to invalidate the 5-factor solution. I would need a more thorough presentation of the rationale. The only one mentioned in the paper is the “relatively small size” of the sample. I guess that if the sample is too small for the regular analysis, it is also too small for 5 different analyses.

-       I’m also worried about the procedure where the author first run a series of CFA, then modify the test to increase fit, then run an EFA…

-       In the discussion, lines 684-, the authors interestingly hypothesize that the high correlation among factors of the GRS could be explained by the global impression a teacher gets from a pupil – a hollow effect. If this is true, then should we consider that the test really measures what it is supposed to? Or is nothing more than an evaluation of the global impression of a teacher? This question is probably worth addressing.

-       On line 705, an article on twice exceptional children is cited to illustrate the tri-partite model. Is that intentional? How relevant is this reference?

-       I feel the discussion could be shortened a little.

-  The general idea that the GRS is in the end correlated with traditional intelligence tests or other cognitive assessments is interesting, but since the paper is on giftedness, I would appreciate to see a translation of this correlation in terms of detection. That is, if a child is gifted when in the top 2.3%, how good that this overlap with the 2,3% higher rates according to teachers?

Author Response

Response to Reviewer 2

Comments

Dear Reviewer thank you very much for all of your comments. You helped us to clarify the content of our work. For this reason, we modify the Introduction according to the suggested reference.

Point 1: Not being native myself, I cannot assess the language, but I found some sentences odd and got the feeling that the paper should be edited. I hope a native reviewer will be able to invalidate or confirm my impression.

-

Answer to Comment 1: Thank you very much for your comments. The manuscript was checked by an English teacher (native speaker also).

Point 2:   I found the theoretical framework unclear. After reading the Introduction carefully, the reader still has no idea about what exactly is the diversity of giftedness conceptions stated by the authors, or what definition the authors embrace. For this reason, many statement are hard to understand. For instance, it is clear that the authors consider the definition of giftedness by high intelligence (g) only as obsolete. However, they still use intelligence to argue about the (convergent) validity of the GRS-P scale. The theoretical framework should be clarified.

Answer to Comment 2: Thank you very much for your comment. The Introduction was rewritten in an attempt to clarify the issue of the assessment of giftedness. New articles were incorporated. (You can track all the changes wich are highlighted with yellow color)

Point 3: References are sometimes irrelevant: papers on specific topics are cited as examples of general conceptual framework, where one would expect a literature review or a theoretical paper.

Answer to Comment 3: Thank you very much for your comments. All the new references are highlighted with yellow.

Point 4:   I’m not familiar enough with all the statistics methods used in this paper to be positive. However, I found that the statistical procedures for study 1 are at least suspicious. For instance, running an EFA after a CFA on the same data sounds odd. Moreover, if my understanding is correct, the scale was revised between the CFA and the EFA, which looks like an unconventional procedure.

Answer to Comment 4: Thank you very much for your comment. We would like to thank the reviewer for giving us the opportunity to clarify the order of presentation of the structural equation modeling (SEM) analyses that we conducted. As we noted in the submitted paper (please see first paragraph in RESULTS) regarding the sample size requirements, for SEM techniques, it is recommended as a rule of thumb that there be at least five observations per estimated parameter (Hair, Anderson, Tatham, & Black,  1998). Given that the GRS-P consists of five scales (namely: Intellectual, Academic, Creativity, Artistic, and Motivation) with 12 items each (60 items), the sample size for Confirmatory Factor Analysis at the item level, concluding the 60 items of the 5 scales, had to exceed 300 observations. Thus, taken into account the relatively small size of the sample, we did not conduct a total CFA concluding 60 items. Alternatively, we conducted a set of five separate confirmatory factor analyses for the data collected from the 12 items that constitute each of the five GRS-P scales and we verified the one-factor structure of each of the five scales of the GRS-P for the Greek kindergarten teachers sample of Study 1. The sample size for each of the 5 Confirmatory Factor Analyses (including the 12 items of each of the 5 scales) had to exceed 60.

It should be noted that the aforementioned set of CFA at the item-level -although they fully verified the proposed by Pfeiffer and Jarosewich (2003) uni-factorial structure for each of the five scales of the Greek version of the GRS-P– were limited as regards the identification of the number of the GRS-P organization’s underlying factors. Therefore, as we noted in the submitted paper (please see the second paragraph in results), since a uni-factorial structure for each of the five scales of the Greek version of the GRS-P was verified (at the item-level data) and taking into account the finding of Benson and Kranzler (2017) at the scale-level data (not at the item-level data), that a general factor (latent variable) has been found to account for most of the variance captured by the five GRS-P ratings (measured variables), we decided to follow Benson’s and Kranzler’s (2017) methodology. In specific, we applied both exploratory factor analysis (EFA) and confirmatory factor analysis (CFA) (at the scale-level) to all five scales of the Greek version of the GRS-P in order for us to be able to identify the number of their organization’s underlying factors (latent variables). These scale-level analyses (both EFA and CFA) were not run at the item-level but at the total scores for the aforementioned verified factor structure of each scale of GRS-P. Thus, the Intellectual Ability, Academic Ability, Creativity, Artistic Talent, and Motivation, scales were treated as observed variables, first in EFA and then in CFA that conducted at the scale-level data.

Point 5: It would be nice to have a more detailed description of the multidimensional model underlying the GRS (or the Munich Model)

Answer to Comment 5: Thank you for your comment. A brief detailed description is presented for the Munich model (you can track the change with yellow)

Point 6: While reading this section the reader needs to know (1) what definition of giftedness and which framework the authors endorse and (2) is the distinction between those using IQ tests and those relying mostly on parents and teachers’ questionnaires merely methodological, or does it correspond to a deep theoretical opposition?

Answer to comment 6. Thank you for your comments. The Introduction is rewritten in an attempt to enhance the assessment of gifted children. All the new references are highlighted also with yellow.

Point 7: Answer to Comment 7: Thank you very much for your Comment: The sentence: “The above indices are extremely high and could suggest redundancy in the items.” has been added at the Discussion section. (please see the second paragraph in the Discussion section

Point 8:  In the discussion, lines 684-, the authors interestingly hypothesize that the high correlation among factors of the GRS could be explained by the global impression a teacher gets from a pupil – a hollow effect. If this is true, then should we consider that the test really measures what it is supposed to? Or is nothing more than an evaluation of the global impression of a teacher? This question is probably worth addressing.

Answer to Comment 8: Thank you very much for your comment. The discussion section is closely aligned with the finding of Psfeiffer 2021, Siou 2010, Li & Pfeiffer 2008, Benson & Kranzler 2018 etc

Author Response

Dear reviewer thank you very much for your comments. All the changes have been highlighted with yellow

Round 2

Reviewer 1 Report

Comments were addressed. Thanks

Reviewer 2 Report

Thanks for the revised version of the paper. I feel that my comments have been addressed adequately.

Reviewer 3 Report

I have read the authors' responses and can be satisfied and recommend its publication.